# Evaluation of Systemic Treatment Options for Gastrointestinal Stromal Tumours

**DOI:** 10.3390/cancers15164081

**Published:** 2023-08-13

**Authors:** Marin Golčić, Robin L. Jones, Paul Huang, Andrea Napolitano

**Affiliations:** 1Department of Radiotherapy and Oncology, Clinical Hospital Center Rijeka, Krešimirova 42, 51000 Rijeka, Croatia; 2Sarcoma Unit, The Royal Marsden NHS Foundation Trust, Fulham Road, London SW3 6JJ, UK; 3Division of Molecular Pathology, The Institute of Cancer Research, Sutton SM2 5NG, UK; paul.huang@icr.ac.uk

**Keywords:** avapritinib, GIST, imatinib, personalised care, regorafenib, ripretinib, sunitinib, targeted therapy

## Abstract

**Simple Summary:**

This review summarises the systemic treatment options available for patients with gastrointestinal stromal tumours (GIST). While surgical treatment is recommended for most localised GIST, pre- or post-operative therapy with imatinib is indicated in patients with a high risk of disease recurrence. For most patients with inoperable or metastatic GIST, imatinib is the first-line therapy. Sunitinib, regorafenib, and ripretinib are licensed as second-, third-, and fourth-line therapy, respectively. However, patients with GIST harbouring specific mutations could be imatinib-resistant and follow different therapeutic schemes. This review evaluates potential medication options for each line of systemic treatment and examines the possibility of personalised treatment. The focus is placed on the tumour mutational profile, treatment-related adverse effects, and patient characteristics. Finally, a multidisciplinary approach is crucial, as combining systemic therapy with surgery, radiotherapy, interventional radiology, and radionuclide therapy can improve outcome.

**Abstract:**

Gastrointestinal stromal tumours (GIST) are the most common mesenchymal tumours of the gastrointestinal tract. Surgical treatment is recommended for the majority of localised GIST, while systemic treatment is the cornerstone of management for metastatic or unresectable disease. While a three-year regimen of imatinib is the standard of care in the adjuvant setting, there is no precise recommendation for the duration of neoadjuvant treatment, where imatinib is usually given between 4 and 12 months. Continuous treatment with imatinib at a dose of 400 mg once per day is recommended for most patients with unresectable or metastatic GIST in the first line. An exception is represented by patients with tumours harbouring the imatinib-insensitive *PDGFRA* D842V mutation who would be better treated with avapritinib. Targeted therapies are also recommended in the presence of NTRK rearrangements and BRAF mutations, although limited data are available. While an increase in the dose of imatinib to 800 mg is an option for the second line, sunitinib is usually considered the standard of care. Similar outcomes were reported for ripretinib in patients with tumours harbouring KIT exon 11 mutation, with significantly fewer side effects. Regorafenib and ripretinib are the standards of care in the third and fourth lines, respectively. The recent development of various systemic treatment options allows for a more personalised approach based on the molecular profile of the GIST, patient characteristics, and the profile of medications’ adverse events. A multidisciplinary approach is paramount since combining systemic treatment with locoregional treatment options and supportive care is vital for long-term survival.

## 1. Introduction

Gastrointestinal stromal tumours (GIST) are the most common mesenchymal tumours of the gastrointestinal tract originating from the interstitial cells of Cajal [1,2]. The histopathological diagnosis relies on both the morphology and the immunohistochemistry markers, such as CD117, the c-kit proto-oncogene product (KIT), and DOG1, which are positive in the vast majority of GISTs (85%) [3]. 

Additional evaluation of genetic mutations further provides both prognostic and predictive value and is especially important in CD117/DOG1 negative GIST. *KIT* mutations are the most common, especially in KIT exon 11 and 9 regions (75%), followed by the mutations in platelet-derived growth factor receptor alpha polypeptide (*PDGFRA*) (10%) (Figure 1). Within the *PDGFRA*, the most frequent mutations are the substitutions of aspartic acid with valine in exon 18 (D842V) [4]. *KIT* and *PDGRFA* mutations are generally considered mutually exclusive. Patients without a mutation in *KIT/PDGFRA* (10–15%) can present alterations in other genes, some of which are targetable (e.g., *BRAF* or *NTRK*), possibly in the context of cancer predisposition syndromes (e.g., *SHD*, or *NF1*) [4,5,6,7,8]. Along with genetic mutations, mitotic index, tumour rupture, size, and location also have important prognostic value and impact treatment recommendations [8,9]. The location of the tumour is of particular value as it often correlates with the mutational status. Proximal gastric GIST is almost exclusively *KIT* mutant (96%), while *PDGFRA* and succinate dehydrogenase (*SDH*)-deficient GIST occurs most frequently in the distal stomach area [10]. Similarly, the prevalence of *KIT* exon 9 mutations is higher in extra-gastric GIST [11].

The diagnosis and treatment of the GIST should be approached by a multidisciplinary team and in hospitals with expertise in mesenchymal tumours. Following the histopathological diagnosis via endoscopy or biopsy, imaging with either computerised tomography (CT), magnetic resonance (MR), or positron emission tomography (PET)/CT is necessary in most cases. Surgical treatment is recommended for most localised GIST, while systemic treatment is the basis for metastatic or unresectable disease [8,9].

Although GISTs are relatively rare tumours, various successful systemic treatment options exist. This manuscript aims to evaluate the medications used for GIST and the potential to optimise and personalise the treatment in adjuvant, neoadjuvant, and metastatic settings.

## 2. Adjuvant Treatment

While the primary treatment of localised GIST is surgical, adjuvant systemic treatment is often necessary due to the high chance of recurrence. The use of imatinib, a selective inhibitor of the KIT kinase receptor (tyrosine-kinase inhibitor (TKI)), is based on a randomised trial comparing imatinib to placebo in patients with a resected GIST >3 cm. One year of adjuvant imatinib resulted in a longer relapse-free survival (RFS) (1 year: 98% vs. 83%, HR 0.35, *p* < 0.0001) but did not improve overall survival (OS) (1 year: 99.2% vs. 99.7%, HR 0.66, *p* = 0.47) [12]. Further analysis of the trial showed that patients with tumours harbouring *KIT* exon 11 deletions derived greater benefits in RFS than patients with a *KIT* 11 insertion or point mutation, *KIT* exon 9 mutation, *PDGFRA* mutation, or wild-type GIST [13]. 

Casali et al. evaluated 2 years of adjuvant imatinib in localised, high- or intermediate-risk GIST. Similar to the previous trial, patients on imatinib derived benefit in RFS (5-year RFS: 69% vs. 63%, *p* < 0.001) but not OS, compared to placebo (5-year OS: 91.8% vs. 92.7%, *p* > 0.05). However, there was a non-significant trend to longer survival in patients with high-risk GIST, which included patients with tumours >10 cm in size, >10 mitoses per 50 high-power fields (HPF), or >5 cm and >5 mitoses per 50 HPF [14]. 

Finally, the current standard of care was established following a trial by Joensuu et al. which included only high-risk patients (as previously defined by Casali et al. [14]) and patients with tumour rupture before or after surgery. A 3-year imatinib regimen resulted in longer RFS (5-year: 65.6% vs. 47.9%, *p* < 0.001, HR 0.46) and OS (5-year: 92.0% vs. 81.7%, *p* = 0.02, HR 0.45) compared to the 1-year regimen [15]. While patients with *KIT* exon 11 mutated tumours have a higher chance of relapse after treatment with curative intention [16], the same group of patients derived the highest benefit of the 3-year treatment (RFS HR 0.35 (0.22–0.56)), compared to patients with *KIT* exon 9 mutated tumours (HR 0.61, 0.22–1.68), no mutations (HR 0.41, 0.11–1.51), or with other detectable mutations (HR 0.78, 0.22–2.78) [15]. Despite the efficacy of imatinib 800 mg dose on *KIT* exon 9 mutated tumours in the advanced setting, higher-dose adjuvant therapy is not recommended since no prospective data are available. This is also supported by a retrospective analysis of a large cohort of patients with resected *KIT* exon 9 mutated GIST [17]. Furthermore, current data suggest that adjuvant therapy with imatinib should be avoided in patients with *NF1*, *NTRK*, *BRAF*, and *PDGFRA* D842V-mutated GIST due to lack of efficacy [8,9,18].

Longer follow-up confirmed the value of 3-year adjuvant imatinib, with a 10-year OS of 79.0% vs. 65.3% (HR 0.55, 0.37–0.83, *p* = 0.004) for the 1-year schedule, even though a higher number of secondary cancers were registered in the 3-year treatment group (17.2% vs. 12.1%). The benefit of the longer imatinib regimen was most pronounced for patients with tumours harbouring *KIT* exon 11 mutations [19]. While the benefit of 3-year imatinib was found regardless of the age, tumour site, tumour rupture, completeness of the surgery, or size, additional analyses showed that only patients with more than 10 mitoses per 50 HPFs derived benefit (HR 0.29, 0.17–0.49). The tolerability remained an issue, with just over one-quarter of patients (25.8%) in the 3-year regimen discontinuing imatinib due to side effects [19,20].

The success of the 3-year regimen has prompted further studies. The PERSIST-5 phase 2 clinical trial evaluated 5 years of adjuvant imatinib in patients with an intermediate or high risk of recurrence. The results showed an estimated 5-year RFS of 90% (95% CI, 80–95%) and OS of 95% (95% CI, 86–99%). However, forty-five patients (49%) stopped treatment early, most commonly due to patient choice or adverse effects [21]. Another retrospective study compared the effect of 5-year vs. 3-year adjuvant imatinib on ruptured localised GIST. A longer course of imatinib resulted in a longer 5-year RFS (78% vs. 30%, *p* = 0.042) but a similar 5-year OS (100% vs. 92.1%, *p* > 0.05) compared to a 3-year regimen. Furthermore, the study confirmed low mitotic index as a significant independent favourable prognostic factor for RFS and confirmed a high discontinuation rate in the 5-year imatinib group (40%, N = 8) [22].

A phase 3 trial (SSG XII) studying three versus five years adjuvant imatinib in high-risk GIST patients is currently underway and should help clarify the optimal length of adjuvant treatment [23].

## 3. Neoadjuvant Treatment

Although imatinib proved successful in the adjuvant setting, only a handful of trials evaluated imatinib in the neoadjuvant treatment of GIST. BFR14 was a phase III trial that included patients with non-metastatic locally advanced primary treated with imatinib 400 mg daily, which was applied both in the neoadjuvant and adjuvant settings. A partial response was achieved in 60% of the patients (N = 15), and 36% underwent surgery (N = 9), leading to a 3-year progression-free survival (PFS) of 67% and OS of 89% [24]. On the other hand, ACCRIN 6665/RTOG 013, a phase II study, evaluated imatinib in now atypical 600 mg daily dose in GIST patients with either primary disease >5 cm or in a metastatic/recurrent setting with disease burden >2 cm. Patients were started on neoadjuvant therapy with imatinib, which continued two years post-operatively. The results showed a response in less than 10% of the patients, with an estimated 2-year PFS of 77–83% depending on the initial metastatic status. The majority of patients without metastatic disease were operated on with R0 resections (77%, N = 20), compared to 58% of patients with metastatic/recurrent GIST (N = 11) [25]. A phase II study in Asian GIST patients with gastric tumours >10 cm evaluated 6–9 months of neoadjuvant imatinib 400 mg daily and achieved a response rate (RR) of 62%, with 91% of the patients achieving an R0 resection (48/53) [26].

Despite a scarcity of neoadjuvant trials, guidelines recommend initiating neoadjuvant therapy in localised GIST with imatinib-sensitive mutations, primarily when surgical morbidity could be reduced with the tumour downstaging. The optimal length of neoadjuvant treatment is not yet elucidated, but treatment is rarely given longer than 12 months as late regressions are rare, and drug resistance might occur. Imatinib can be stopped prior to surgery and resumed as soon as the patient can tolerate oral medications, which is compared favourably to other TKIs used in GIST, where at least a one-week pause before surgery is recommended [8,9].

The follow-up in the treatment of GIST can be challenging as there is a lack of biomarkers that can be used to evaluate treatment response, although emerging data evaluating the presence of *PDGRFA* and *KIT* mutations in circulating DNA (ctDNA) showed that certain mutations can predict clinical benefit from different types of TKIs [27,28,29]. 

## 4. First-Line Treatment of Metastatic or Unresectable GIST

### 4.1. Imatinib 

Treatment with imatinib has also revolutionised the prognosis of patients with metastatic or unresectable GIST. Although complete responses are rare, a partial response can be observed in over half of the patients, and stable disease is achieved in over a quarter [30]. Imatinib is most frequently administered in a 400 mg daily dose since long-term follow-up showed no difference in survival between 400 mg, 600 mg, or 800 mg doses in unselected GIST patients; the median OS of the whole patient group was 3.9 years, with 10-year OS up to 21.5% [31]. However, imatinib plasma exposure seems to play a critical role in its effectiveness, as patients with a trough concentration of less than 1100 ng/mL exhibited a shorter PFS and a lower rate of clinical benefit [32].

Similar to the experiences from the adjuvant setting, mutational status significantly impacts the success of treatment in metastatic GIST. Patients with *KIT* exon 11 mutations exhibited significantly higher odds of achieving a response compared with patients with *KIT* exon 9 mutations (odds ratio (OR), 4.35; 95% CI, 2.44 to 7.96), wild-type *KIT* (OR, 11.1; 95% CI, 5.88 to 20), or other mutations (OR, 3.13; 95% CI, 1.31 to 7.69 [29]. While SDH deficiency and *PDGFRA* D842V mutations resulted in primary resistance to imatinib [33,34], the relative resistance observed in *KIT* exon 9 mutations could be overcome by increasing the imatinib dose to 800 mg [35]. 

Imatinib treatment interruptions should be avoided if possible, and the treatment for metastatic or unresectable disease should be continuous, as imatinib cessation in patients with stable GIST after either 3 or 5 years was associated with a higher rate of progression [36,37]. Furthermore, a lower quality of response might be achieved after the reintroduction of imatinib [38]. Even for patients with refractory GIST, imatinib cessation resulted in a flare phenomenon, suggesting that imatinib might still be effective for a subset of GIST cells [39].

While imatinib resulted in excellent disease control in the majority of patients, early resistance was noted in up to 13.6% (N = 20) of patients [30], and only 9.5% of patients were progression-free at 10 years [31], suggesting that further advances are needed and that combining locoregional treatments with systemic therapy might be crucial for long-term survival [40,41,42].

### 4.2. Avapritinib

In patients with *PDGFRA* D842V mutations, treatment with imatinib is not recommended due to ineffectiveness. However, the recently published NAVIGATOR trial evaluated avapritinib, a novel *KIT* and *PDGFRA* inhibitor, on adult patients with unresectable GIST, of which 11 patients (20%) were treatment naïve. A prespecified focus was placed on *PDGFRA* D842V-mutant GIST. Despite being a phase I study, the results were unprecedented, with an overall RR of 91% (51/56 patients) and a median PFS of 34.0 months (95% CI: 22.9-NR). The median OS was not reached during the follow-up [43,44]. Due to the exceptional results of the NAVIGATOR trial, both NCCN and ESMO guidelines recommend avapritinib as a first-line therapy for GIST patients with D842V mutation [8,9]. 

### 4.3. Other Therapy Options 

Both guidelines also support the use of *NTRK* inhibitors in the first-line setting if a targetable mutation is found due to the success of tumour-agnostic trials, which reached a RR of 75% (95% CI 61–85) for larotrectinib (which included three patients with GIST) and 57% (95% CI 43.2–70.8) for entrectinib, which included a single patient with GIST [45,46]. *BRAF* inhibitors are also a potential option in metastatic or unresectable GIST, although there is a scarcity of data on their efficacy in metastatic GIST [8,9,47].

Several other medications have also been tested in a first-line unresectable or metastatic GIST setting. Dasatinib, a short-acting inhibitor of multiple tyrosine kinases including *KIT* and *PDGFRA*, has been evaluated in TKI naïve patients with metastatic GIST in continuous 2 × 70 mg dose, exhibiting a promising RR of 74% (31/42 patients, 95% CI 56–85%) based on the FDG-PET/CT follow up at 4 weeks and a median PFS of 13.6 months [48]. Masitinib, a TKI with a greater in vitro selectivity for the wild-type c-Kit receptor compared to imatinib, was tested in a phase II study in the first-line setting at 7.5 mg/kg daily dose, resulting in the RR of 20% (6/30 patients) when evaluated by RECIST criteria but 86% (12/14 patients) when using the FDG-PET response criteria. Median time-to-response was 5.6 months (95% CI 0.8–23.8 months), median PFS was 41.3 months, and a 3-year OS was 89.9% (95% CI 71.8; 96.6) [49].

## 5. Second-Line Treatment of Metastatic or Unresectable GIST

### 5.1. Increasing the Dose of Imatinib

The choice of second-line treatment of progressive GIST depends on various factors, including the presence of imatinib-sensitive mutations and the choice of first-line medication. Patients who have started and progressed on imatinib have two main potential options for further treatment. The first includes the increase of the dose of imatinib to 800 mg daily, as up to 33% of patients who crossed over to the high-dose imatinib regimen achieved a clinical benefit. However, an elevated imatinib dosage schedule resulted in a higher percentage of grade 3–5 toxicities, which could hamper patient compliance and treatment efficiency [8,50].

### 5.2. Sunitinib

Another option includes switching the treatment to sunitinib, a multitargeted TKI, following the results of a phase III trial comparing sunitinib against a placebo upon progression to imatinib. The study showed that sunitinib in a daily dose of 50 mg in 6-week cycles (4 weeks on, 2 weeks off treatment) resulted in a longer PFS compared to placebo (27.3 weeks (95% CI 16.0–32.1) vs. 6.4 weeks (4.4–10.0) (HR 0.33; *p* < 0.0001), respectively) [51]. A longer follow-up, which allowed for a cross-over using a rank-preserving structural failure time method, reported an OS of 72.7 vs. 39.0 weeks (HR 0.505, 95% CI 0.26–1.13; *p* = 0.306) [52]. Similar to imatinib, the treatment efficacy of sunitinib appears to be dependent on the type of mutations present in the GIST. Patients with *KIT* exon 9 mutation or wild-type GIST exhibited significantly longer PFS than patients with exon 11 mutations [51]. Furthermore, sunitinib also seems to be efficacious in SDH-deficient GIST with an RR of up to 18.4% (7/38 patients), which is important as imatinib is less efficient in GIST with those molecular subtypes [53,54]. While most studies evaluated sunitinib in a 4/2 scheme for GIST patients, various other dosing schedules were also tested. A phase II study in GIST patients showed that continuous dosing of 37.5 mg daily sunitinib results in an acceptable RR and safety profile, while morning versus evening dosing did not affect the efficacy or safety profile [55]. While not tested in GIST patients, there is an abundance of data in renal cancer showing the benefit of a 2/1 scheme of sunitinib. Meta-analysis showed a superior PFS, fewer adverse events, and fewer treatment interruptions with a 2/1 scheme compared to a 4/2 scheme [56].

### 5.3. Other Tyrosine-Kinase Inhibitors

Although increasing the imatinib dose is a valid option upon progression, most trials have tested different medications primarily against sunitinib in the second-line setting. The phase III trial INTRIGUE compared ripretinib with sunitinib. There was no difference in PFS for GIST patients in the intention-to-treat (ITT) population or with *KIT* exon 11 mutation (HR 0.88, 95% CI 0.66–1.16; *p* = 0.36; median 8.3 vs. 7.0 months), while OS data were not yet mature for the analysis. However, patients on ripretinib exhibited higher RR in the KIT exon 11 ITT population (HR 23.9% (39/163) vs. 14.6% (24/164), *p* = 0.03) and a significantly lower number of grade 3 and 4 side effects (41.3% (N = 92) vs. 65.6% (N= 145), *p* < 0.0001). The results suggest that ripretinib might be an option for selected patients with *KIT* exon 11 mutation [57].

In a phase II trial, sunitinib was also tested against masitinib upon disease progression to imatinib. Patients treated with masitinib reported severe adverse events less frequently (52% (12/23) versus 91% (19/21), *p* = 0.008) and exhibited a longer OS compared to sunitinib (29.8 months (95% CI 17.8–NR) vs. 17.4 months (95% CI 9.4–28.6), *p* = 0.033). No difference was found for PFS between the two arms (3.7 months (95% CI 1.9–6.0) vs. 1.9 months (95% CI 1.8–4.4, *p* = 0.833). Data from a phase III trial are expected in the coming years [58].

Other agents were also evaluated upon disease progression on imatinib but without a standard control arm. Dasatinib, in 2 × 70 mg daily dose, was evaluated in the second line and beyond and achieved a 6-month PFS rate of 29% and a RR of 25% (N = 12), including a patient with *PDGFRA* exon 18 mutation, which is resistant to the treatment with imatinib [59]. 

Additionally, a phase II trial evaluated linsitinib, an oral TKI with specificity for insulin-like growth factor-1 receptor, in both pediatric and adult patients with *KIT*/*PDGFRA* wild-type GIST, after progression on at least one treatment line (although the median number of treatments was three). While no objective responses were recorded, PFS and OS at 9 months were 52% and 80%, respectively [60].

### 5.4. Non-Tyrosine-Kinase Inhibitors Options

While TKIs are the most commonly used drug for treating GIST, several trials evaluated medications with different mechanisms. A phase II trial evaluated immune checkpoint inhibitors, either nivolumab or nivolumab with ipilimumab upon disease progression in the second line and beyond; the median was three lines of treatment. A clinical benefit rate was reported for 52.6% of patients on nivolumab (10/19) and 31.3% for patients on combination immunotherapy (5/16). However, one patient (6.7%) treated with the combination achieved a complete response. Treatment with single-agent nivolumab achieved a PFS of 11.7 weeks (95% CI 7.0–17.4), while combination immunotherapy resulted in a median PFS of 8.3 weeks (95% CI 5.6–22.2). However, the primary endpoint of >15% RR was not reached for either treatment arm [61].

A combination schedule of everolimus, an inhibitor of the mammalian target of rapamycin, combined with imatinib was evaluated in GIST patients upon progression on imatinib monotherapy using a dosage scheme of everolimus 2.5 mg with imatinib 600 mg daily. A PFS of 1.9 months and a median OS of 14.9 months were reported [62]. 

Encouraging research was also recently published on temozolomide, an oral alkylating agent often used to treat primary central nervous system tumours, on SDH-deficient GIST models. While temozolomide was only tested on five patients, the RR was 40%, the disease control rate was 100%, and median OS was 6.4 years from diagnosis and 1.9 years from the start of treatment in patients with a median of at least one prior line of TKI. The responses were not dependent on the volume of metastatic disease [63,64].

## 6. Third-Line Treatment of Metastatic or Unresectable GIST

### 6.1. Regorafenib

Regorafenib, a multikinase inhibitor, is usually considered the standard in the third-line setting. Initially, a phase II trial of regorafenib 160 mg daily for 21 days following a 7-day break, upon progression to at least imatinib and sunitinib, achieved a clinical benefit rate of 79% (26/33, 95% CI 61–91%) and a PFS of 10 months [65]. The final follow-up of the same trial resulted in a median PFS of 13.2 months (95% CI 9.2–18.3 months) and an OS of 25 months (95% CI 13.2–39.1 months) [66]. The positive results of this trial led to a successful phase III trial (GRID), which confirmed the efficacy of regorafenib compared to placebo plus the best supportive care. Regorafenib exhibited significantly longer PFS compared with the placebo, although the PFS was shorter than in the initial phase II trial (4.8 months vs. 0.9 months, HR 0.27, 95% CI 0.19–0.39; *p* < 0.0001). Subgroup analysis of the phase III trial has shown that regorafenib was equally efficient in both *KIT* exon 11 and 9 mutated GIST [67]. In contrast, the phase II trial reported significantly better results for *KIT* exon 11 mutations than *KIT*/*PDGFRA* wild-type GIST and suggested a potential benefit of regorafenib in an SDH-deficient setting (PFS of 10 months) [65,66]. The value of regorafenib was also shown for GIST patients with *KIT* exon 17 mutations, generally thought to be insensitive to both imatinib and sunitinib. In this group of pretreated patients, regorafenib resulted in a remarkable PFS of 22.1 months with a clinical benefit rate of 93.3% at 4 months (14/15 patients) [68]. Real-world data confirmed the safety and efficacy of regorafenib in the third-line setting and beyond, with a Korean study showing a PFS of 4.5 months (95% CI 3.8 to 5.3) and OS of 12.9 months (95% CI, 8.1 to 17.7) [69]. However, both GRID and the Korean trial showed a requirement for dose modification in 72–77% of patients, leading researchers to evaluate different dose regimens [67,69]. Kim et al. tested the efficacy and safety of regorafenib at a lower dose and on a continuous dosing schedule (100 mg once orally daily) and reported a median PFS of 7.3 months (95% CI 5.9–8.6), and the 1-year OS of 64.5%, while only 24% of patients (6/25) required a dose modification [70]. Experience using regorafenib in colorectal cancer has also shown the potential of dose-escalating treatment, resulting in significantly fewer side effects [71].

### 6.2. Avapritinib 

The VOYAGER trial recently evaluated avapritinib in 300 mg dose versus regorafenib in the third-line or later in patients with unresectable or metastatic GIST. However, there were no significant differences in PFS between the two drugs (4.2 vs. 5.6 months; *p* = 0.055), with similar disease control rates (41.7% (95% CI, 35.4 to 48.2) and 46.2% (95% CI 39.7 to 52.8)) [72]. However, as the NAVIGATOR trial also showed [43,44], avapritinib was effective in *PDGFRA* D842V–mutant GIST with significantly higher PFS (median not reached (NR), 95% CI 9.7 to NR vs. 4.5 months; 95% CI 1.7 to NR; *p* = 0.035) and disease control rate (100.0%, 7/7 patients (95% CI 59.0 to 100.0) versus 33.3%, 2/6 patients (95% CI 4.3 to 77.7)) compared to patients on regorafenib [72].

### 6.3. Rechallenge with Imatinib

Imatinib rechallenge in the third-line or beyond in patients with metastatic or unresectable GIST patients who previously experienced response or stable disease longer than 6 months was evaluated by a phase III trial (RIGHT). Compared to placebo, rechallenge with imatinib resulted in a longer PFS (1.8 months (95% CI 1.7–3.6) versus 0.9 months (95% CI 0.9–1.7); HR for progression or death was 0.46 (95% CI 0.27–0.78, *p* = 0.005). While the results were modest, as reintroduction showed the possibility of slowing but not halting disease progression, imatinib rechallenge might be a viable option, especially in countries with limited drug availability [73]. Real-world data also support imatinib rechallenge in the absence of other alternative treatment options [74].

### 6.4. Other Treatment Options 

Nilotinib was evaluated after imatinib and sunitinib progression; initially, a single-arm phase 2 study reported a median PFS of 113 days (3.7 months; 90% Cl, 102–119 days), with a median OS of 310 days. However, while the RR rate was 3% (1/35), one of the responses was recorded on a patient with imatinib-resistant and sunitinib-resistant KIT exon 17 mutation [75]. Nilotinib was also compared to the best supportive care with or without another TKI in a phase III trial on GIST patients after progression on at least two lines of treatment. There were no differences in the PFS (109 days versus 111 days, *p* = 0.56) or OS (332 days versus 280 days, *p* = 0.29), although a high level of discordance was reported between local and central reviews. Furthermore, a post-hoc analysis, including only patients in the third-line setting, reported a significantly longer OS for nilotinib (405 versus 280 days, HR = 0.67, 95% CI (0.48, 0.95); *p* = 0.02) compared to the best supportive care. The safety profile was also acceptable, with 15.7% of grade 3–4 side effects (N = 39) compared to 12.0% in the control group (N = 10) [76].

Various TKIs have also been tested in third-line or beyond but not in a phase III setting. Pazopanib was evaluated in GIST patients who had disease progression on both imatinib and sunitinib by a phase II trial PAGIST, resulting in a PFS of 19.6 weeks (95% CI 12.6–23.4) [77]. An open-label phase II trial (PAZOGIST) was conducted in a similar setting and achieved a similar median PFS of 3.4 months (95% CI 2.4–5.6) for patients treated with pazopanib, which was significantly longer compared only to best supportive care (2.3 months (95% CI 2.1–3.3), HR 0.59 (95% CI 0.37–0.96), *p* = 0.03) [78].

A phase II trial (CaboGIST) evaluated cabozantinib after progression on imatinib and sunitinib in GIST patients, resulting in a median PFS of 5.5 months (95% CI 3.6–6.9), RR of 14% (N = 7), and a DCR of 82% (N = 41). However, over half of the patients required dose interruptions due to side effects (54%, N = 27) [79].

Dasatinib was prospectively evaluated in a phase II study as the third-line treatment in Asian patients, with a dose escalation scheme (starting with 50 mg twice daily, escalating to 70 mg twice daily after the first week). The trial reported a RR of 3.4% (2/58), a median PFS of 3.1 months (95% CI, 2.77–3.23 months), while a median OS was 14.0 months (95% CI, 11.89–16.1 months). Interestingly, neither primary nor secondary gene mutations predicted the efficacy of dasatinib. At the same time, 31.0% of patients required treatment interruptions (N = 18) [80]. As the trials in earlier lines also showed, dasatinib might be particularly valuable in treating imatinib-resistant *PDGFRA* D842 mutations [59,80].

Sorafenib was also tested exclusively on Asian patients with metastatic GIST who were in the third line or beyond. Sorafenib was administered orally at 400 mg twice daily and resulted in an RR of 13% (N = 4) (95% CI 1–25%), PFS of 4.9 months (95% CI 1.3–8.5 months), and OS of 9.7 months (95% CI 7.2–12.2 months). A small subset of patients who were previously treated with nilotinib in the third-line setting before sorafenib or who had a GIST mutation other than KIT exon 11, exhibited worse disease control compared to other patients. A total of 32.2% (N = 10) of patients required dose interruption or reduction, most commonly due to hand–foot syndrome [81].

## 7. Fourth-Line Treatment of Metastatic or Unresectable GIST

### 7.1. Ripretinib

While many of the drugs previously mentioned in the second- or third-line setting of metastatic or unresectable GIST allowed the inclusion of patients in the fourth-line setting, several trials evaluated exclusively patients in the fourth-line or beyond. The current standard of care in the fourth line was established following the results of the INVICTUS trial, which compared ripretinib to placebo. Ripretinib, a switch-control TKI active against a broad spectrum of *KIT* and *PDGFRA* mutations, was tested in a 150 mg dose in GIST patients upon progression to three other TKIs. The median PFS was 6.3 months (95% CI 4.6–6.9) for ripretinib and 1.0 months (95% CI 0.9–1.7) for placebo (HR 0.15, 95% CI 0.09–0.25; *p* < 0.0001). Ripretinib also resulted in a longer OS compared to placebo (15.1 months (95% CI 12.3–15.1) versus 6.6 months (95% CI 4.1–11.6), HR 0.36, 95% CI 0.21–0.62). Treatment with ripretinib exhibited a trend toward a higher RR (9.4% (8/85) versus 0%, *p* = 0.0504), with an excellent safety profile as only 9.4% (8/85) of the patient on ripretinib reported severe side effects compared to 6.9% (3/43) of patients on placebo. An additional value of ripretinib is its effectiveness in *KIT* and *PGDFRA* wild-type GIST patients [82]. Ripretinib also resulted in a higher quality of life compared to the placebo (nominal *p* < 0.01) [83]. For a subset of patients who progressed in the INVICTUS trial, a dose escalation of ripretinib was allowed (2 × 150 mg daily dose). The follow-up resulted in a median PFS, from randomisation to progressive disease, of 4.6 months (95% CI 2.7–6.4), while the median OS was 18.4 months (95% CI 14.5-not estimable). Similar to the original trial, the safety profile was acceptable, with abdominal pain being the most common grade 3–4 side effect (5%). A total of 26% (N = 11) of patients required treatment interruption, and 16% discontinued the treatment (N = 7) [84].

### 7.2. Pimitespib

The only other phase III trial in the same setting compared pimitespib, a heat shock protein 90 inhibitor, in a 160 mg daily dose to a placebo. The trial (CHAPTER-GIST-301), which was conducted only in Japan, resulted in a significantly longer PFS of pimitespib (2.8 months (95% CI 1.6–2.9) vs. 1.4 months (0.9–1.8), HR 0.51 (95% CI 0.30–0.87) (*p* = 0.006)). Results of the OS, after cross-over adjusting, confirmed the improvement of OS for patients on pimitespib compared with placebo (HR 0.42, 95% CI 0.21–0.85, *p* = 0.007). The beneficial results on survival were irrespective of the type of *KIT* mutation. Furthermore, only 5.2% (3/58) of patients discontinued the treatment, and only 10.3% (6/58) reported treatment-related serious adverse events. However, a rare side effect of night blindness was reported in 13.8% of patients treated with pimitespib (8/58) [85].

### 7.3. Other Treatment Options

While avapritinib was allowed in the various lines of treatment in the original NAVIGATOR trial, including the first line of treatment [43,44], an analysis of Chinese patients who most commonly received avapritinib in the fourth line of treatment or beyond (34%) resulted in a similar efficacy, with a clinical benefit rate of 86% (24/28) in patients with *PDGFRA* D842V mutation and a PFS of 5.6 months, with immature OS data [86]. A network meta-analysis, which has evaluated the efficacy and safety of third-line and over-third-line therapy, has placed ripretinib as the best drug for PFS, OS, and disease control rate, whereas nilotinib and pimitespib were shown to have more tolerable side-effects [87].

## 8. Discussion 

For most patients with metastatic or unresectable GIST, the treatment schedule is clear for the first four lines: imatinib, sunitinib, regorafenib, and ripretinib. However, advances in the molecular understanding of GIST, evaluation of head-to-head studies, and research emphasising the importance of various patient characteristics have allowed us to evaluate particular patient subgroups where a different treatment plan should be followed. 

### 8.1. Mutation Profile 

In patients with *PDGFRA* D842V mutations, avapritinib should be used as a first-line of treatment. While guidelines do not suggest a precise second-line treatment, enzyme assays demonstrated the efficacy of ripretinib against D842V mutation [88], and a trial evaluating dasatinib achieved a RR of 25%, which included a patient with *PDGFRA* exon 18 D842V mutation [59]; those medications could also be an option where avapritinib is not readily available. 

Patients without a mutation in the *KIT* or *PDGFRA* could harbour a variety of other mutations, some of which could be targetable. SDH-deficient GIST is also resistant to imatinib, and sunitinib could be considered due to an RR of 18.4% [53,54]. Following sunitinib, regorafenib is usually suggested as it achieved a PFS of 10 months in this patient subgroup [65,66]. Interesting results were also shown for temozolomide, which resulted in an RR of 40%, although on only five patients with SDH deficiency, suggesting the need for further trials [63,64]. For patients with *NTRK* rearrangements or *BRAF* mutations, guidelines allow first-line treatment with *NTRK* or *BRAF* inhibitors [8,9]. However, despite excellent results in tumour-agnostic settings, only a handful of GIST patients were included in those trials. 

Several treatment options exist for patients with *KIT*/*PDGFRA* wild-type GIST (no targetable mutation). Dasatinib exhibited a 71% RR in patients with *KIT*/*PDGFRA* wild-type GIST in TKI-naïve patients [48], while wild-type GIST patients treated with sunitinib in the second-line or beyond exhibited longer PFS compared to patients with *KIT* exon 11 mutated tumours [51]. Following sunitinib, treatment with regorafenib could be a potential option for those patients [65]. While the results of the INVICTUS trial could not confirm the efficacy of ripretinib in *KIT*/*PDGFRA* wild-type GIST in the fourth line or beyond, the preclinical analysis does show potential efficacy [89]. Furthermore, data on paediatric and adult patients have shown that linsitinib may have activity in this group of patients [60]. 

### 8.2. Quality of Life and Adverse Effects 

While the treatment choice based on the type of mutation is crucial, quality of life and specific adverse events could also have a role in decision making, especially when considering data from large head-to-head studies. For example, the INTRIGUE trial compared ripretinib to sunitinib in the second-line setting. While the two medications showed similar efficacy in *KIT* exon 11-mutated patients in terms of PFS (HR 0.88, *p* = 0.36), ripretinib exhibited a significantly lower number of severe side effects, both leading to a lower percentage of treatment interruptions (29.1% vs. 41.6%) and discontinuation (3.6% vs. 7.7%). Furthermore, the adverse events profile differed between the drugs, with neutropenia, thrombocytopenia, dysgeusia, and stomatitis almost exclusive to sunitinib, while alopecia and seborrheic keratosis dominating the ripretinib arm [57].

Similar results in the same setting were obtained for masitinib versus sunitinib, although from a phase II trial. While the two medications exhibited a similar PFS (HR 1.1, *p* = 0.833), patients on masitinib reported a significantly lower number of severe side effects and a trend toward lower treatment discontinuation (4% (1/23) vs. 24% (5/21), *p* = 0.088). The side-effect profile was also different, with a significantly higher percentage of nausea/vomiting (70% vs. 33%, *p* = 0.033) reported in the masitinib arm compared to abdominal pain (4% vs. 33%, *p* = 0.19), thrombocytopaenia (0% vs. 33%, *p* = 0.003), hypertension (4% vs. 33%, *p* = 0.019), mucosal inflammation, dysgeusia, and hand–foot syndrome (all 4% vs. 29%, *p* = 0.042) in the sunitinib arm [58]. 

In the third-line setting, avapritinib and regorafenib have exhibited similar PFS and disease control rates. The percentage of discontinuation (8.3% vs. 5.6%) and serious adverse effects were similar (19.7% vs. 14.5%), although the adverse-effect profile differed. Grade 3 or higher anaemia (20.9% vs. 2.6%), leukopenia (4.2% vs. <1%), and cognitive impairment of any grade (25.9% vs. 3.8%) were more common in the avapritinib arm. On the other hand, grade 3 or higher diarrhoea (1.7% vs. 6.8%) and hypertension (1.7% vs. 12.0%) were more often present in regorafenib [72].

While there are no head-to-head trials in the fourth line, a specific profile of adverse events of ripretinib (alopecia) and pimitespib (night blindness) was also reported [84,85]. A meta-analysis suggested nilotinib and pimitespib as the treatment with the most tolerable side-effect profile in the third line or beyond [87].

Hence, as the evaluated medications from these studies suggest a similar efficacy for the selected patient population, the adverse-event data allows for personalising the treatment, especially considering patient comorbidities and preferences. Evaluating adverse effects in the treatment decision is crucial since data show that while most studies reported maintained the quality of life during the TKI treatment, reported side effects could have been underestimated by physicians, as shown by a high discontinuation rate shown by many of the trials [90]. Further head-to-head studies, especially those including the non-TKI, such as everolimus, temozolomide, nivolumab, and ipilimumab, could further broaden the treatment choice. 

### 8.3. Patient Characteristics 

Along with consideration of the presence of specific GIST mutations and the difference in the safety profile of the systemic treatment options, a variety of patient-related factors should be considered. For example, patients who undergo major gastrectomy were reported to have significantly lower levels of imatinib in plasma [91], previously associated with a worse therapeutic response [92]. However, the exact mechanism that results in lower imatinib concentration is unknown and was not associated with the level of the acidic environment; imatinib exposure did not increase normal levels when exposed to a more acidic environment [93]. On the other hand, gastrectomy did not seem to affect exposure to sunitinib [94], regorafenib [95], or ripretinib [96].

While the initial chart review suggested shorter survival of patients treated with sunitinib and acid-suppressing agents [97], additional research showed discordant results [98]. Similarly, the lack of effect of pantoprazole was noted for the pharmacokinetics of ripretinib [99] and imatinib [100]. On the contrary, omeprazole significantly decreased the area-under-curve and Cmax of dasatinib [101], suggesting caution when combining acid-suppressing agents with particular TKIs. 

Furthermore, patients’ nutritional status is a factor that should also be addressed during systemic treatment. Data have shown that 77.8% of GIST patients were at risk of malnutrition, with an incidence of malnutrition of 10.1%, with GIST location and size as independent risk factors for nutritional status [102]. While not tested on GIST patients, research has shown that low pre-treatment serum albumin, a marker of malnutrition, is an independent adverse predictor of the prognosis of renal cancer patients receiving TKI therapy [103]. Hence, to optimise systemic treatment, it is also critical to tackle malnutrition. However, a proper diet must be chosen carefully, as research on rodents has shown that a high-salt diet during sunitinib treatment is associated with increased blood pressure and glomerular injury [104]. 

A growing body of evidence shows the gut microbiome’s importance for the success of checkpoint immunotherapy, which is also used in GIST [61]. While there is a lack of data on the importance of the microbiome in TKI metabolism, Zimmermann et al. [105] showed that, in vitro, human gut bacteria could metabolise various drugs, including imatinib (affected by *Parabacteroides johnsonii*, *Bacteroides eggerthii*, *Bacteroides vulgatus*, and *Bacteroides stercoris*) and dasatinib (metabolised by *Clostridium bolteae*, *Bifidobacterium ruminatum*, and *Bacteroides fragilis*). An association was also established between diarrhoea during the sunitinib treatment in renal cancer patients and the decrease in butyrate-producing bacteria and the increase of *Bacteroidetes* [106]. It was also shown that sunitinib itself could induce a significant shift in the microbiome that could affect further treatment [107]. Recent data has also shown that tissue microbial composition of GIST and microGIST exhibit a difference in the abundance of various bacteria, such as the enrichment of Proteobacteria in GIST samples, hypothesising that the microbiome restructuration can drive the carcinogenesis process [108]. While microbiome studies are scarce in the GIST, the microbiome likely plays a significant role during the treatment with TKI or immunotherapy. Future studies could lead to further treatment optimisation, including the potential use of probiotics or faecal microbial transplantation. 

### 8.4. The Potential of Radiotherapy and Radionuclide Treatment

Every treatment decision in metastatic GIST patients should be based upon a review of the multidisciplinary team in a hospital with experience in treating the GIST. However, the experience with systemic treatment has shown that only a minority of patients with metastatic GIST will be disease-free at 10 years. Hence, new systemic treatments, along with locoregional treatments, are paramount in achieving long-term success [31,40,41,42]. However, new treatment options, including radiotherapy and radionuclide therapy, are under consideration for GIST patients. The trial MITIGATE-NeoBOMB1 has evaluated the safety of 68Ga-NeoBOMB1, a gastrin-releasing peptide receptor antagonist, in GIST patients. The results exhibited a promising safety profile and pharmacokinetics, which could help open an entirely new treatment avenue [109]. 

Along with radionuclide therapy, stereotactic ablation radiotherapy has shown tremendous potential in treating oligometastatic disease when combined with systemic therapy. Although GIST is not generally thought to be radiosensitive, studies have shown that radiotherapy can be used successfully to relieve symptoms of GIST of advanced or metastatic lesions and help achieve an objective response without reducing the quality of life [110,111]. 

Additionally, radiofrequency ablation and other interventional radiology techniques can be used for the treatment of solitary progression (particularly liver metastases) of sarcoma and GIST patients on tyrosine kinase inhibitor therapy [112,113]. This can enable patients to continue on aTKI for longer. 

## 9. Conclusions and Future Directions

While GIST is a rare disease, systemic treatment options are relatively abundant and efficacious (Table 1), although not all of the discussed drugs are readily accessible to patients internationally. However, mutational status, adverse events, and patient characteristics must be carefully considered when choosing the treatment strategy. Additionally, complementary therapy, including surgery, radiotherapy, and radiofrequency ablation, could all be helpful in order to optimise the treatment for patients with GIST but must be evaluated by a multidisciplinary team on a case-to-case basis in hospitals with experience in treating sarcomas. 

A growing field of research is also emphasising the importance of ctDNA, which could potentially reduce the need for obtaining an often hard-to-get tissue biopsy and identify the subgroups of patients who could obtain the highest benefit with a particular treatment schedule. Furthermore, despite the advances in survival with the application of modern therapies, there is a need for a greater understanding of *KIT*/*PDGFRA* wild-type GIST and the optimal therapeutical sequences in those patients. 

The treatment landscape for metastatic GIST is likely to change over the next few years. Post-hoc analysis of the INTRIGUE trial showed that ripretinib had better disease control compared to sunitinib in patients with ctDNA harbouring *KIT* exon 11 and co-occurring *KIT* exon 17 and/or 18 mutations, whereas patients with *KIT* exon 11 and 13/14 mutations had better outcomes with sunitinib [29]. This has led to the second line INSIGHT trial, randomising patients in this molecular subgroup to receive either ripretinib or sunitinib [116]. INSIGHT represents the first clinical trial in GIST with patients selected based on their mutational profile on ctDNA, and the results of this trial are eagerly awaited. In addition, the current PEAK trial randomises patients in the second-line setting to receive sunitinib alone or the combination of sunitinib and bezuclastinib as second-line therapy for advanced GIST [117]. This trial highlights the potential of combination therapy in advanced GIST.

In addition, there are a number of very promising agents in development, including novel TKIs such as IDRX-42, THE-640, and NB003 [118,119,120]. Consequently, the treatment of advanced GIST is likely to evolve further with greater precision in management.

## Figures and Tables

**Figure 1 cancers-15-04081-f001:**
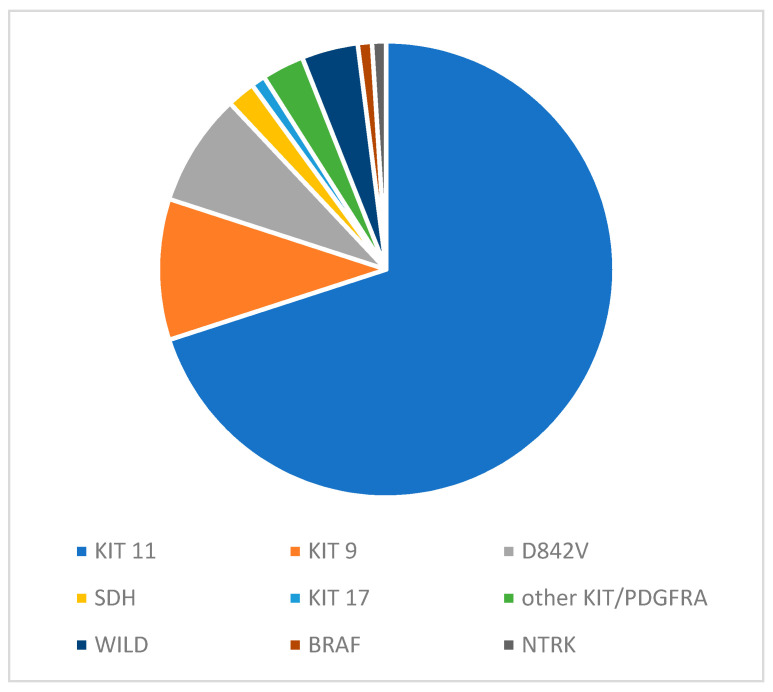
An approximation of the relative incidence of GIST mutations in the unresectable or metastatic setting.

**Table 1 cancers-15-04081-t001:** An overview of randomised clinical trials in unresectable or metastatic GIST.

Investigational Medication	Earliest Line of Treatment	Dose	Comparator	Response Rates	Survival Analysis
Imatinib [30]	1st line	400 mg daily	Imatinib 600 mg	49.3% vs. 58.1% (*p* > 0.05)	PFS, OS n/a
Imatinib [30,114]	1st line	400 mg daily	Imatinib 800 mg	50.1% vs. 54.3% (*p* > 0.05) Updated: 51.0% vs. 56.7% (*p* > 0.05)	PFS n/a, 0.82 (*p* = 0.026) 1-year OS: 85% vs. 86% (*p* > 0.05) Updated PFS 1.7 vs. 2.0 years (HR 0.91, *p* > 0.05) Updated OS 3.9 vs. 3.9 years (HR 0.93, *p* > 0.05)
Imatinib [50]	1st line	400 mg daily	Imatinib 800 mg	44.6% vs. 45.8% (*p* > 0.05)	PFS 18 vs. 20 months (HR n/a, *p* = 0.13) OS 55 vs. 51 months (HR 0.98, *p* = 0.83)
Nilotinib [115]	1st line	800 mg	Imatinib 400 mg	42.3% vs. 51.9% (*p* = n/a)	2-year PFS 51.6% vs. 59.2% (HR 1.47, *p* < 0.05) 2-year OS 81.8% vs. 90.0% (HR 1.85, *p* < 0.05)
Sunitinib [50]	2nd line	50 mg (4/2 scheme)	placebo	7% vs. 0% (*p* = 0.006)	PFS 27.3 vs. 6.4 weeks (HR 0.33, *p* < 0.0001) OS n/a, HR 0.49, *p* = 0.007)
Ripretinib [57]	2nd line	150 mg daily	sunitinib	21.7% vs. 17.6% (*p* = 0.27) KIT11 ITT 23.9% vs. 14.6% (*p* = 0.03)	PFS 10.6 vs. 10.3 months (HR 0.88, *p* = 0.81) PFS *KIT11* ITT 8.3 vs. 7.0 months (HR 0.88, *p* = 0.36), OS n/a
Masitinib [58]	2nd line	12 mg/kg daily	sunitinib	n/a	PFS 3.7 vs. 1.9 months (HR 1.1, *p* = 0.833) OS 29.8 vs. 17.4 months (HR 0.40, *p* = 0.033)
Nivolumab [61]	2nd line	240 mg every 2 weeks	Nivo-lumab + ipili-mumab	0% vs. 1% (*p* > 0.05)	PFS 11.7 vs. 8.3 weeks (HR n/a, *p* = 0.99) OS 26.9 vs. 8.8 months (HR n/a, *p* = 0.19)
Regorafenib [67]	3rd line	160 mg daily D1-21/28	placebo	4.5% vs. 1.5% (*p* > 0.05)	PFS 4.8 vs. 0.9 months (HR 0.27, *p* < 0.0001) OS n/a (HR 0.77)
Avapritinib [72]	3rd line	300 mg daily	regorafenib	17.1% vs. 7.2% (*p* < 0.001)	PFS 4.2 vs. 5.6 months (HR 1.25, *p* = 0.055) 1-year OS estimate: 68.2% vs. 67.4% (HR n/a, *p* > 0.05)
Imatinib [73] (rechallenge)	3rd line	400 mg daily	Placebo	0% vs. 0% (n/a)	PFS 1.8 vs. 0.9 months (HR 0.46, *p* = 0.005) OS 8.2 vs. 7.5 months (HR 1.00, *p* = 0.92)
Pazopanib [78]	3rd line	800 mg daily	Best supportive care	0% vs. 1% (*p* > 0.05)	PFS 3.4 vs. 2.3 months (HR 0.59, *p* = 0.03) OS 17.8 vs. 12.9 months (HR 0.94, *p* = 0.69)
Ripretinib [83]	4th line	150 mg daily	placebo	9.4% vs. 0% (*p* = 0.0504)	PFS 6.3 vs. 1.0 months (HR 0.15, *p* < 0.0001) OS 15.1 vs. 6.6 months (HR 0.36)
Pimitespib [85]	4th line	160 mg daily	Placebo	1.7% vs. 0% (*p* > 0.05)	PFS 2.8 vs. 1.4 months (HR 0.51), OS 13.8 vs. 7.6 months (HR 0.42)

ITT = intention-to-treat population. n/a = not available. OS = overall survival. PFS = progression-free survival.

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
