# Peer review of "Evaluation of Systemic Treatment Options for Gastrointestinal Stromal Tumours"

_cancers, 2023, doi:10.3390/cancers15164081_

Round 1
Reviewer 1 Report
Thank you for asking me to review this paper. This is a very well researched, excellent review of the management of GIST and the authors should be congratulated on writing an excellent paper. The paper takes the reader through the management of GISTs both in the adjuvant and metastatic settings and the evolution of treatment options with the increasing use of mutational analysis as a guide. I have learnt a lot from reading this and I would strongly support publication.
The abstract is a lovely summary of the paper. A minor change would be that it might be better to change the wording of the sentence ....PDGFRA D842V mutations who should be...to PDGRFRA D842V mutations who would be better treated with avapritinib to make it less commanding. In addition, is it worth specifically mentioning exon 9 mutations for doubling the dose of imatinib in the metastatic setting here?
I appreciate that this paper is to be read internationally but is it worth mentioning that not all the drugs discussed are readily accessible to all patients in all countries? A discussion of what to do if they are not available would be helpful, for example for the D842V patients who can't access avapritinib.
On page 4, line 161 there is a typo where prior to surgery reads priori.
I do not think any other changes need to be made.
Overall I think this is a brilliant up-to-date summary of the systemic therapy management of GISTs and I would recommend publication.
Author Response
Dear Reviewer,
Thank you for your kind words; we have changed the Abstract as per your suggestions; while we agree it would be worth further expanding the abstract, we are limited by the maximum number of words. We have added a sentence in the conclusion, noting the potential lack of accessibility of the drugs internationally, and added a comment regarding the potential lack of avapritinib availability in 8.1 paragraph.
Kind regards.

Reviewer 2 Report
The authors present a clear and comprehensive overview of the evolution and current-state-of-affairs regarding treatment and management of gastrointestinal stromal tumors (GIST), particularly locally advanced or metastatic GISTs. This is a really nice and readable manuscript for anyone, clinicians and researchers alike, interested in GIST. The review refers to the relevant literature and clearly highlights the different therapeutic approaches needed for GISTs with specific mutations. Importantly, it also stresses the need for experienced multidisciplinary teams to make treatment decisions. In these respects GIST may be exemplary for other cancers treated with targeted small molecules. I have only a few minor comments.
Minor Comments:
1. Page 14, line 621 – “patient” should be plural so “patients”, please check and correct.
2. Page 14, line 630 – 632 – It would be interesting to indicate the targets of the agents (i.e. IDRX-42, THE-640 and NB003) that are in development.
Author Response
Dear Reviewer,
Thank you for your comments. The minor comments have been addressed in the appropriate lines of text.
Kind regards
